# (Un)reasonable Allure of Ante-hoc Interpretability for High-stakes Domains: Transparency Is Necessary but Insufficient for Comprehensibility

**Kacper Sokol** [1]   **Julia E. Vogt** [1]

## Abstract

Ante-hoc interpretability has become the holy grail of explainable artificial intelligence for high-stakes domains such as healthcare; however, this notion is elusive, lacks a widely-accepted definition and depends on the operational context. It can refer to predictive models whose structure adheres to domain-specific constraints, or ones that are inherently transparent. The latter conceptualisation assumes observers who judge this quality, whereas the former presupposes them to have technical and domain expertise (thus alienating other groups of explainees). Additionally, the distinction between ante-hoc interpretability and the less desirable post-hoc explainability, which refers to methods that construct a separate explanatory model, is vague given that transparent predictive models may still require (post-)processing to yield suitable explanatory insights. Ante-hoc interpretability is thus an overloaded concept that comprises a range of implicit properties, which we unpack in this paper to better understand what is needed for its safe adoption across high-stakes domains. To this end, we outline modelling and explaining desiderata that allow us to navigate its distinct realisations in view of the envisaged application and audience.

## 1. Unpacking Ante-hoc Interpretability

Data-driven predictive models are often classified as either *transparent* (i.e., comprehensible) or *black-box*. The second label is assigned to systems that are *proprietary*, hence inaccessible, or *too complicated* to understand [Rudin, 2019]. Setting the former situation aside, the latter presupposes an observer who appraises the intelligibility of the model in question. This suggests that interpretability may not be a binary property, but it is rather determined by the degree to which one can understand the system's operations, and thus resides on a *spectrum of opaqueness* [Sokol & Flach, 2021]. While ultimately a model may be deemed comprehensible or black-box, this judgement will differ across observers. In view of this fluidity, the distinction between *ante-* and *post-hoc* techniques commonly made in the literature invites further scrutiny.

Post-hoc explainability offers insights into the behaviour of a predictive system by building an additional *explanatory model* that interfaces between it and human explainees (i.e., explanation recipients). This is the only approach compatible with black-box systems, with which we can interact exclusively by manipulating their inputs and observing their outputs. Such an operationalisation is nonetheless unsuitable for high-stakes domains, e.g., medicine or healthcare, since the resulting explanations are not guaranteed to be reliable and truthful with respect to the underlying model [Rudin, 2019]. Ante-hoc interpretability, in contrast, often refers to data-driven systems whose model form adheres (in practice) to application- and domain-specific constraints – a *functional* definition – which allows their designers to judge their trustworthiness and communicate how they operate to others [Rudin, 2019].

While the post-hoc moniker is, in general, used consistently across the literature, the same is not true for the ante-hoc label. The latter sometimes refers to, among other things, transparent models that are self-explanatory, systems that are inherently interpretable, scenarios where the explanation and the prediction come from the same model, and situations where the model itself (or its part) constitutes an explanation, which are vague terms that are often incompatible with the functional definition of ante-hoc interpretability. Nonetheless, all of these notions presuppose an observer to whom the model is interpretable and who understands its functioning, which brings us back to the issue of accounting for an audience and its background.

The functional definition of ante-hoc interpretability concerns the engineering aspects of information modelling and its adherence to domain-specific knowledge and constraints. While these properties are of paramount importance for safe adoption of predictive systems in high-stakes appli-

---

[1]Department of Computer Science, ETH Zurich, Switzerland. Correspondence to: Kacper Sokol <kacper.sokol@inf.ethz.ch>.

*Workshop on Interpretable ML in Healthcare at International Conference on Machine Learning (ICML)*, Honolulu, Hawaii, USA. 2023. Copyright 2023 by the author(s).

cations by ensuring reliable, robust and trustworthy modelling that is open to scrutiny, they neglect the needs and expectations of more diverse audiences. A model may be ante-hoc interpretable to its engineers and domain experts with relevant technical knowledge, yet it may not engender understanding in explainees outside this milieu. To complicate matters further, explainability is a domain-specific notion that cannot be encapsulated by one definition [Rudin, 2019; Sokol & Flach, 2021]. Ante-hoc interpretability is therefore a multi-faceted concept requiring a human-centred and context-aware approach that allows the designers of explainable artificial intelligence (XAI) systems to account for the degree to which a model should be understood across distinct populations [Miller, 2019].

Explainees have become the key consideration when designing XAI systems, making it pertinent to ask "Explainable to whom and in what situation?" before choosing a suitable technique [Keenan & Sokol, 2023]. Such deliberations are commonplace with respect to the type of an explanation – e.g., counterfactual or feature importance – its presentation medium – e.g., textual or visual – and the complexity of information it communicates – e.g., human-understandable features or their unintelligible embeddings [Small et al., 2023]. They are also relevant to post-hoc methods since each tool can construct an explanation of an arbitrary disposition given that it operates independently of a predictive model. For ante-hoc interpretability, however, such considerations are limited since we tend to lack the agency to adapt an explanation to a particular scenario given that it is sourced directly from a predictive model. This state of affairs curtails the range of explainees who can understand how the predictive model in question operates, possibly impeding the adoption of inherently transparent data-driven systems. Some of these problems arise because *ante-hoc interpretability* is an ambiguous and overloaded concept that, due to its context-dependent nature, lacks a precise and widely-accepted definition, leading to a multitude of its subjective conceptualisations.

To address the aforementioned issues and support adoption of ante-hoc interpretability in high-stakes domains, we investigate the implicit set of properties encompassed by this notion. First, in Section 2, we explore the characterisation of and distinction between ante- and post-hoc XAI techniques. We show, among other things, that inherent transparency of a model may not suffice to engender understanding in explainees, which can necessitate additional processing of the model's structure, thus blurring the line separating these two types of XAI approaches. For example, a decision tree is transparent, but its interpretability depends on its small size (given technical understanding of its operation and familiarity with the data domain); to restore interpretability to a large tree we can process its structure to extract parsimonious decision rules.

Next, in Section 3, we uncover two sets of properties that are fundamental to ante-hoc interpretability:

**modelling desiderata**  span characteristics expected of predictive models (Section 3.1); and

**explaining desiderata**  pertain to the process responsible for extracting the desired explanatory insights from those models (Section 3.2).

The former deal with how (inherently transparent) predictive models process information and the degree to which the components of this pipeline are human-intelligible; the latter concern sources and modularity of explanatory information, the mechanisms employed to extract it, auxiliary or background knowledge needed to interpret it, and human or algorithmic reasoning applied to it to construct understanding. A visual summary of this framework is shown in Figure 1.

Navigating these properties allows us to arrive at different conceptualisations of ante-hoc interpretability. Their correct mixture can help us to build an XAI system that generates explanations suitable for the envisaged audience and application. In this position paper, we refrain from making specific recommendations along these dimensions and instead offer an insightful and thought-provoking discussion about ante-hoc interpretability and what is expected of it in different explanatory contexts (given their plurality). Composing such a prescriptive list would be misguided or deceptive at best and counterproductive at worst in view of our multi-faceted conceptualisation of ante-hoc interpretability. We examine these aspects, summarise our findings and conclude in Section 4.

## 2. Separating Ante- and Post-hoc XAI

XAI lacks a widely-accepted nomenclature and coherent definitions of common terms such as transparency, explainability and interpretability [Sokol & Flach, 2021]. Auxiliary concepts, including the post- and ante-hoc monikers, are also used inconsistently, possibly leading to confusion. While the former often applies to explanations extracted from an (independent) explanatory model built atop an (opaque) predictive system, the latter is more ambiguous. It can refer to inherently transparent models, which implicitly evokes human explainees, or to the functional definition of interpretability, which relies on a constrained model form, technical knowledge and domain expertise. The second conceptualisation aligns with *algorithmic transparency* and *decomposability* but forgoes *simulatability* [Lipton, 2018]; the former two require individual elements of the machine learning process – input data, learning algorithm, model parameters and prediction computation – to be comprehensible, whereas the third warrants manual replication of the

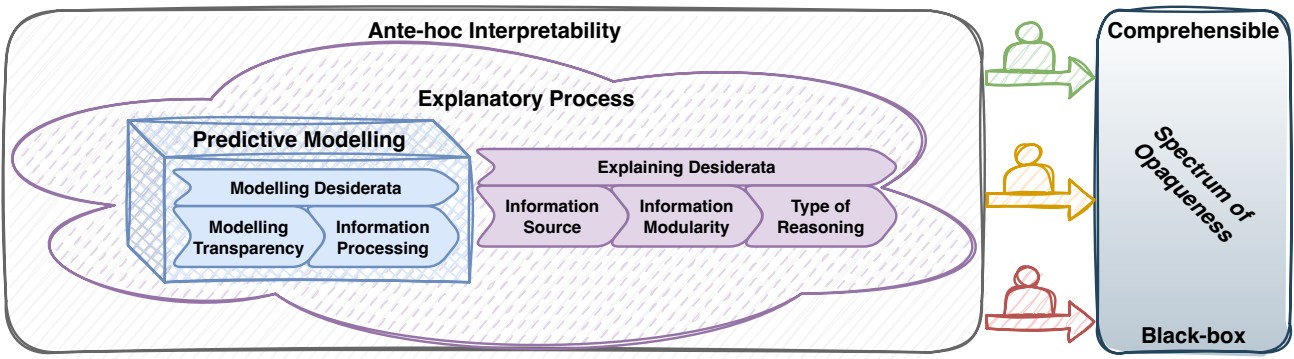

*Figure 1.* Ante-hoc interpretability is an overloaded and multi-faceted concept that implicitly encompasses various *predictive modelling* desiderata (transparency and information processing) as well as properties pertaining to the employed *explanatory mechanism* (information source, information modularity and type of reasoning). A combination of these factors determines a specific realisation of the ante-hoc interpretability paradigm and places it on the spectrum of opaqueness based on its operational context, which encompasses the envisaged use case and audience, among many other factors.

model's operation.

While the functional definition of ante-hoc interpretability entails stronger, and often necessary for high-stakes applications, requirements, both conceptualisations depend on an audience, who either remains unspecified or is presupposed to have relevant technical and domain knowledge. This ambiguity blurs, to a degree, the line separating ante- and post-hoc methods; the explainees are expected to be sufficiently skilled to *reason* about, or interpret, the transparent insights offered by (the structure of) a predictive model – a (post-factum) process that may as well be done by a deterministic, trustworthy and reliable algorithm employed to generate truthful explanations. Delegating this task to an explainer makes the distinction between the two types of XAI techniques less clear while at the same time offering an opportunity to democratise the reach and appeal of ante-hoc interpretability beyond audiences with technical and domain expertise.

While such a scenario has been recognised on multiple occasions, its ramifications are largely overlooked. Rudin [2019] briefly notes that some inherently transparent models may lose their interpretability when becoming overwhelmingly large, however their explanations can remain small by only showing relevant fragments of said models to explainees (whenever such an action preserves truthfulness of these insights). Sokol & Flach [2021] define explainability as:

$$\text{Explainability} \; = \; \underbrace{\text{Reasoning} \left( \text{Transparency} \mid \text{Context} \right)}_{understanding},$$

which elucidates the role of human or algorithmic reasoning, i.e., interpreting, applied to the insights provided by a transparent model in view of its operational context, which spans the deployment scenario and explainees' background knowledge, among other factors. The relation between these three components highlights the reliance on, possibly external, information that is part of the anticipated expertise. Specifically, the reasoning may be purely *procedural* – where the explainee is expected to just work with the insights offered by the system – or it may be *creative* – where auxiliary knowledge (and ingenuity) are needed to understand the model's operation.

For example, consider an inherently transparent sparse linear model that has been trained on complex feature embeddings, thus forgoing its interpretability, and with any form of reasoning offering little help. Counterfactual explanations illustrate a different aspect of our argument; while they tend to be generated post-hoc, i.e., by a separate algorithm that is independent of the underlying model, they are truthful with respect to its predictive behaviour, albeit they only offer a partial and incomplete view that does not generalise to other cases (unless a sufficient number of explanations and a suitable reasoning mechanism are provided). Similarly, individual conditional expectation – which visualises the response of a model when a selected feature of a data point is varied [Goldstein et al., 2015] – is generated post-hoc yet truthful to said model. Revisiting the example of an overwhelmingly large tree that is inherently transparent but no longer interpretable, its explainability can be restored with either algorithmic or human reasoning. Instead of generating a counterfactual in a model-agnostic fashion, we can leverage access to the model's internal structure and compose such an explanation based on any two adjacent leaves [Sokol, 2021]; other forms of structural explainability are also possible.

# 3. Ante-hoc Interpretability Desiderata

The discussion and examples presented in the previous section illustrate the multi-faceted nature of ante-hoc interpretability. To help better navigate this complex landscape we introduce a principled, property-based framework – summarised in Figure 1 – that spans desiderata pertaining to the explained predictive model (Section 3.1) and the interpretability strategy employed to extract relevant explanatory insights (Section 3.2). These come together to offer distinct realisations of ante-hoc interpretability, the recognition of which allows engineers to better fulfil the needs and expectations of diverse explainees.

## 3.1. Modelling Desiderata

The first set of desiderata deals with the predictive modelling aspect of ante-hoc interpretability. This perspective covers technical properties of data-driven models in view of their comprehensibility.

**Are Modelling Components Transparent?**   The basic tenet of ante-hoc interpretability is the transparency of the data modelling process. This is in contrast to post-hoc explainability, which is often employed when a model relies on an unintelligible or inaccessible data representation, thus requiring a (human-interpretable) proxy to generate explanatory insights [Sokol & Flach, 2020b], which process may not capture the model's true behaviour [Rudin, 2019]. This facet is commonly discussed in the literature [Lipton, 2018; Rudin, 2019; Sokol & Flach, 2020a] – as Section 2 has shown – and given its fundamental role we include it for completeness. It concerns the transparency and human-comprehensibility of the elements constituting the machine learning process: data and their features as well as trained models and their operation that maps inputs to predictions.

All of these properties should be considered in view of the *anticipated audience* to avoid *unintelligible transparency* since explanatory mechanisms intrinsic to inherently interpretable models may be misleading to selected groups of explainees. For example, Bell et al. [2022] reported that when asked to identify the most important feature based on a visualisation of a decision tree structure, users tend to select the attribute used by the root node of the tree, which is not necessarily correct. Engineering incomprehensible features, e.g., to improve performance, or relying on predictive models that internally transform understandable attributes into complex representations harms this aspect of ante-hoc interpretability.

**How Is Information Processed?**   How a model handles information, specifically the input features and their mapping to predictions, also affects its perceived comprehensibility. The two broad types of information processing are

*sequential* and *concurrent*, with the latter further characterised by the *operational independence* of processing with respect to pieces of information or collections thereof. An example of a model that processes data sequentially is a decision tree, which selectively and progressively narrows down the scope of consideration by filtering, compartmentalising and contextualising information, thus offering insights that are parsimonious enough to be human-understandable. Neural networks, in contrast, process data concurrently, thereby preventing their observers from grasping meaningful insights into how inputs are transformed and mapped to predictions. The intrinsic structural hierarchy of their architecture and operational independence of layers found therein can nonetheless offer us a glimpse into their functioning, e.g., by visualising (human-intelligible) patterns and concepts learnt by individual filters [Bau et al., 2017; Olah et al., 2018].

## 3.2. Explaining Desiderata

Transparency of a predictive system, as outlined above, is a prerequisite for understanding the operation of its parts or as a whole. This property by itself, however, is insufficient for comprehensibility as we still need a mechanism to extract, reason about and interpret relevant insights into such models – a process characterised by the desiderata outlined below. This aspect of explainable artificial intelligence – the chasm between transparent modelling and explainee understanding – is largely overlooked in the literature [Keenan & Sokol, 2023]. Addressing this sociotechnical gap is likely to require insights from *human cognition* to study explanatory human–machine interactions; specifically, how a diverse collection of explanations and the (interactive) process through which they are delivered affect the user's ability to develop the desired level of understanding [Mueller et al., 2021].

**Is Interpretability Modular?**   Small transparent models, such as shallow trees or sparse linear classifiers, that can be grasped in their entirety are inherently interpretable given relevant technical and domain knowledge. As their size increases they may nonetheless become overwhelming, especially if their structure or information processing strategy prevents their modular interpretation. Models that process data hierarchically, e.g., classification and regression trees [Breiman et al., 1984], or whose predictions depend on factors that can be handled independently, e.g., generalised additive models [Hastie & Tibshirani, 1986], can with certain caveats remain interpretable regardless of their complexity. Presenting parsimonious and cognitively digestible chunks of information may in some cases require reducing the scope of an explanation and abstracting away certain aspects of the model's operational context, thus, while remaining truthful, implicitly offering a partial and incomplete view. For example, counterfactuals are appealing because

they prescribe a small alteration to one's situation – a necessity that limits them to a single instance and requires other features to remain unchanged; a large decision tree may not be interpretable as a whole, but most of its structure can be ignored when investigating a certain subspace or when one logical rule is responsible for the predictive behaviour in question; or a linear model may rely on a sizeable number of features, but its additive structure allows studying the influence of their selected subset.

In addition to controlling for explanation size, which is a purely technical consideration, we must also account for human cognition. People can only hold $7 \pm 2$ objects in their working (short-term) memory [Miller, 1956]; however, a repeated exposure to novel pieces of information allows them to forge new *chunks of knowledge* that can be readily retrieved from long-term memory [Larkin et al., 1980]. This process streamlines cognition by allowing people to map freshly perceived objects onto pre-existing concepts held in their mental model. XAI research in this space is limited, with the most relevant contribution studying the effect of three types of explanation complexity on its utility: explanation size, creation of new types of cognitive chunks and repeated exposure to concepts across explanations [Lage et al., 2019].

**Where Does Information Come From?** Being truthful with respect to a predictive model does not preclude generating explanatory insights post-hoc as these can be just as faithful, sound and complete as ante-hoc transparency under certain conditions, some of which we have outlined above. The dichotomy between ante- and post-hoc approaches stems from the *provenance* and *lineage* of explanatory insights as opposed to their generation mechanism. Provenance of an explanation determines the source of information it relies on. *Endogenous* insights are extracted directly from a (transparent) model and operate on the same concepts (features), thus they are guaranteed to be truthful even if they are retrieved post-hoc by an independent algorithm; *exogenous* insights are based on a separate (explanatory) model, e.g., a surrogate, or employ a different data representation [Sokol & Flach, 2020b]. Explanatory information lineage, which is similar to *translucency* [Robnik-Šikonja & Bohanec, 2018], specifies the manner in which a (transparent or explanatory) model contributes information to an explanatory insight. Model (sub)structure, its formulation, parameters or specific (training) data all exhibit high translucency; introspection mechanisms that elucidate selected aspects of the model's behaviour through probing, e.g., manipulating inputs and analysing outputs, are of low translucency.

**Who Does Reasoning?** Once an admissible piece of transparent information is identified, only its correct interpretation warrants genuine understanding of the underlying model. For example, recall the people's tendency to misconstrue the feature appearing in the root split of a decision tree as the most important attribute [Bell et al., 2022]. The reasoning that underpins this process can be *human*, *algorithmic* or *hybrid*, and it implicitly relies on the (technical and domain) expertise, which may alienate some explainee groups. Additionally, selected forms of transparency may require human ingenuity and creativity as well as auxiliary or background knowledge to spark new insights that lead to useful explanations. Examples of different reasoning are interpreting a rule-based model (human), extracting counterfactuals from decision tree leaves (algorithmic), analysing a model response curve generated with individual conditional expectation (hybrid), and interpreting coefficient relationships such as "feature 1 is twice as important as feature 2" shown in lieu of a large linear model (hybrid). In the latter two cases, explainees need to uncover the meaning as well as judge the importance and implications of explanatory insights generated with an (introspective) algorithm.

## 4. Discussion and Conclusion

This paper showed the ambiguous divide between ante- and post-hoc XAI approaches. Since these concepts lack widely-accepted definitions, ante-hoc interpretability may result in inherently transparent models that remain unintelligible due to how they processes information or an overlooked audience mismatch. To engender their understanding in selected groups of explainees, we may additionally need to consider modularity of the explanatory insights, their provenance and lineage, as well as the human or algorithmic reasoning applied to extract and process them. These desiderata span properties pertaining to (inherently transparent) models and techniques used to make sense of them, offering a principled framework that aids in navigating XAI methods. For example, it allows us to precisely describe various types of explainability based on the provenance–lineage spectrum; the functional definition of ante-hoc interpretability can be characterised by high translucency of endogenous explanatory insights, whereas post-hoc methods are broadly understood to have low translucency or use exogenous information.

High-stakes domains, such as healthcare, require the highest level of intelligibility for which the functional notion of ante-hoc interpretability is best suited; however, without extending its specification further, it may fail to achieve the desired goals as demonstrated throughout this paper. While our desiderata may not be exhaustive, they are guaranteed to stimulate insightful discussion and offer a good starting point for their future expansion.

## Acknowledgements

This research was supported by the Hasler Foundation, grant number 21050.

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
