# OpenReview forum: "(Un)reasonable Allure of Ante-hoc Interpretability for High-stakes Domains: Transparency Is Necessary but Insufficient for Comprehensibility"
_ICML.cc/2023/Workshop/IMLH — IMLH 2023 PosterShortPaper_

### Official Review · Reviewer_8L8z · 2023-06-02

**Rating:** 3
**Confidence:** 3

**Review:**

The paper proposes a series of desiderata to steer the conversation regarding Ante-hoc explainability of ML models.
I find the paper needs a bit more work. The paper correctly identifies the confusion and diverging opinions in the field of X-AI but i do not believe it contributes sufficiently to the disambiguation of the ideas in the debate. The paper is also in need of clearer explanations and writing as there are often tangents and parenthetical evidence that break up the flow of the argument.
I will refrain from going into details of the arguments posed by the paper and how they stand in terms of reason, as i believe they are in need of bit more thought and clarity.

Overall, even though an interesting subject i do not believe this is an appropriate venue for this paper at its current form

---

### Official Review · Reviewer_3JEZ · 2023-06-09
**Need to add experiments to prove the conclusion**

**Rating:** 3
**Confidence:** 4

**Review:**

This paper, in the field of explainable machine learning for high-stacks domains, the authors tried to show that Ante-hoc interpretability is an overloaded concept that spans a range of implicit properties. The authors supposed to prove their conclusion in mainly two aspects, model-based-desiderata and explainer-based-desiderata. I think they might need to design some experiments or use quantitative ways to prove their results in a more directly way.

---

### Official Review · Reviewer_P88r · 2023-06-17

**Rating:** 6
**Confidence:** 3

**Review:**

This article takes a deep dive into the concept of ante-hoc interpretability, specifically in high-stakes fields like healthcare. It highlights the challenges and importance of understanding how machine learning models can be explained before they are used. The authors provide valuable insights and offer practical considerations for building transparent and trustworthy AI systems. Overall, it is a valuable resource for those interested in making AI more understandable and reliable.

---

### Meta-Review · Area_Chair_xZtC · 2023-06-19

**Recommendation:** Accept (Poster)
**Confidence:** 5

**Metareview:**

[Short paper]

The perspective paper examines the two concepts of ante-hoc and post-hoc explainability, and argues the term ante-hoc is overloaded and there should be a spectrum rather than the dichotomy in the XAI algorithmic space. The argument is solid with no identified logic flaw. The insightful and thought-provoking discussion is helpful for the XAI community, and the healthcare application may also benefit from the discussion, as the authors argue that the interpretability is domain- or user-specific. To improve the clarity, I suggest the authors adding actionable items to summarize the interpretability suggestions/desiderata regarding modelling and XAI methods, and adding a figure to aid readers' understanding of the arguments, for example, an illustration of the spectrum and the location of each desideratum on this spectrum.

---

### Decision · Program_Chairs · 2023-06-20

Accept (Poster Short Paper)